# Desynchronized liquid crystalline network actuators with deformation reversal capability

Yao-Yu Xiao[1], Zhi-Chao Jiang[1], Jun-Bo Hou[1] & Yue Zhao [1✉]

Liquid crystalline network (LCN) actuator normally deforms upon thermally or optically induced order-disorder phase transition, switching once between two shapes (shape 1 in LC phase and shape 2 in isotropic state) for each stimulation on/off cycle. Herein, we report an LCN actuator that deforms from shape 1 to shape 2 and then reverses the deformation direction to form shape 3 on heating or under light only, thus completing the shape switch twice for one stimulation on/off cycle. The deformation reversal capability is obtained with a monolithic LCN actuator whose two sides are made to start deforming at different temperatures and exerting different reversible strains, by means of asymmetrical crosslinking and/or asymmetrical stretching. This desynchronized actuation strategy offers possibilities in developing light-fueled LCN soft robots. In particular, the multi-stage bidirectional shape change enables multimodal, light-driven locomotion from the same LCN actuator by simply varying the light on/off times.

[1] Département de chimie, Université de Sherbrooke, Sherbrooke, QC J1K 2R1, Canada. ✉email: yue.zhao@usherbrooke.ca

Liquid crystalline networks including elastomers (LCNs) have become a particularly promising material system for intelligent actuators and soft robots, thanks to their ability to reversibly and macroscopically deform upon the order-disorder phase transition, when properly processed[1-10]. By adjusting the alignment of mesogens and/or the distribution of crosslinking domains, monolithic LCNs can display a wide range of pre-designated deformations, including those based on contraction/extension, bending, twisting and their various possible combinations[11-19]. Accordingly, a myriad of complex shapes (e.g., wave, accordion, helix, saddle shapes, periodical patterns, and 3D profiles of Gaussian curvatures)[20-29] as well as robotic and bionic motions (e.g., gripping, rolling, walking, swimming, and oscillating)[30-40] have been achieved, making the field flourish. Up to date, the reversible shape change of LCNs only involves two shapes corresponding to the isotropic state (disordered state) and the LC phase (ordered state), respectively. The shape evolution between these two states is typically unidirectional, which means that, for example, a ring (shape 1 in LC phase) evolves directly into a flat strip (shape 2 in isotropic phase) upon heating (Fig. 1a), generating a series of intermediate shapes in the flattening process. Here, we report a reversible shape change mode of monolithic LCNs containing a reversal of deformation direction upon unidirectional order-disorder or disorder-order phase transition. In contrast to the conventional transition between shape state 1 and 2, the deformation reported here includes an additional deformation stage II with reversed shape evolution direction compared to stage I deformation and a resulting third shape state (shape 3). As an example, a monolithic LCN actuator can transform from the original ring shape (shape 1) to a flat strip (shape 2), and subsequently reverse the course to form a ring shape of different size (shape 3), and all this only by heating (Fig. 1b). Due to the reversibility, the shape change is reversed during cooling—from shape state 3 to 2, and finally back to shape state 1. This reversible actuation behavior that includes a deformation direction reversal in each half of an actuation cycle is referred to as the deformation-reversal behavior in our work.

One may notice that, the reversal of the bending direction under continuous illumination was reported for some photomechanical actuators[41-43]. Generally, when light is on, a flat strip bends first and then flattens. This is known to originate from the formation at short light-on times and the subsequent elimination at prolonged illumination times of a stress gradient along the thickness direction of the strip, which can be induced photochemically or photothermally through the use of photoswitches or photothermal agents in LCNs, respectively. However, in all cases, as the light-induced gradient and its variation over time is essential, when light is off, the strip cannot bend and flatten again, because the cooling is uniform for photothermally-based actuators, or because the photoreaction is ceased for photochemically driven actuators[41,42]. As shown below, this apparent deformation reversal (bending/unbending), which is the intermediate state as the LCN sample moves to the equilibrium state under irradiation, is different from the built-in deformation-reversal capability explored in the present study.

In this study, the reversible deformation-reversal actuation is achieved without the necessity of having a temporal, light-induced stress gradient. We adopt a strategy based on desynchronizing the thermomechanical responses of the two sides of a monolithic LCN by structurally implementing different actuation temperatures and forces, which, we show, can be obtained through regulating the photocrosslinking times and/or draw ratios of the two sides. Since the deformation-reversal behavior stems from structurally inscribed desynchronized actuation, it occurs when the LCN is subjected to uniform heating and can be observed in a thermal equilibrium state. Using this strategy, it is possible to adjust or control the deformation-reversal actuation in terms of the magnitude and the shape-changing directionality. The LCN actuators can also be programmed by photopatterning or through offsetting LC alignment to display wave- or helix-based deformation-reversal behavior under uniform stimulation. All these features are hard, if not impossible, to realize through the formation of a temporal thermomechanical (or photomechanical) gradient. The deformation-reversal capability of LCN actuators, highlighted as an exploitable property in our study, offers possibilities in developing soft robotic applications. As a proof of concept, by taking advantage of the fact that one deformation-reversal actuation cycle can be divided into multiple stages, we prepared LCN actuators that can execute light-driven locomotion in multiple fashions (referred to as multimodal locomotion) by tuning only the light-on/off times for desired temperature variations.

## Results

**Crosslinking-determined mechanical response**. The LCN actuator, capable of executing two-stage shape morphing towards opposite directions on only stimulation on or off, is fabricated based on the concept of "desynchronized actuation". It means that the two sides of a LCN strip start to deform at different temperatures and impose different strains. To validate this strategy, the material used in this work is a main-chain liquid crystalline polymer containing biphenyl groups as mesogens and the cinnamyl groups as photocrosslinkable moieties (Fig. 1c, synthesis procedures, previously reported in ref. [21], are detailed in Supplementary Methods). The liquid crystalline polymer film (0.35 mm in thickness if not otherwise stated) was given a planar, uniaxial alignment of mesogens by mechanical stretching to 400% at 58 °C (LC phase). After being irradiated with UV light (320 nm, 180 mW cm$^{-2}$), the polymer was photocrosslinked through the generated cinnamyl dimers, thereby locking the oriented mesogens for the reversible deformation (a typical dimension of the actuator obtained is 9.5, 0.65, and 0.16 mm for length, width and thickness, respectively). Interestingly, the crosslinking density can affect the LCN's order-disorder phase transition and further influence the actuation behavior. Figure 1d shows the differential scanning calorimetry (DSC) curves of the uniaxially aligned LCN of which both sides were crosslinked for 20, 60, and 120 min, respectively. As the crosslinking time increases, the LC-isotropic phase transition temperature range widens while the peak position shows no significant change. This apparently similar peak maximum may be attributed to the following combined effects: (1) the effect of crosslinking the stretched sample in the LC phase, which favors the macroscopic order and provides a tendency to increase the LC-isotropic phase transition temperature ($T_{LC-I}$), and (2) the effect of disordering the LC order by the crosslinks (acting as defects), which tends to reduce $T_{LC-I}$[44,45]. As for the broadening of the phase transition region, it can be caused by both polymer chain stretching and crosslinking. As previously reported[46], stretching of the LCN can suppress the smectic phase in the polymer and emerge the initial smectic-nematic and nematic-isotropic phase transition peaks into a broad nematic-isotropic peak. Furthermore, increasing the crosslinking degree of unstretched polymer appears to exert a similar effect due to the chain constraints that hinder the smectic layer formation (see more details in Supplementary Fig. 1). The broader phase transition of the more crosslinked LCN corresponds to a wider actuation temperature range, which can be reflected by the thermally induced reversible strain. With the three strips subjected to the same cooling and heating run (3 °C min$^{-1}$), their length changes were recorded using a dynamic mechanical analyzer in the iso-stress mode (Fig. 1e). In all cases, the LCN strip extends on cooling into the LC phase and then contracts on

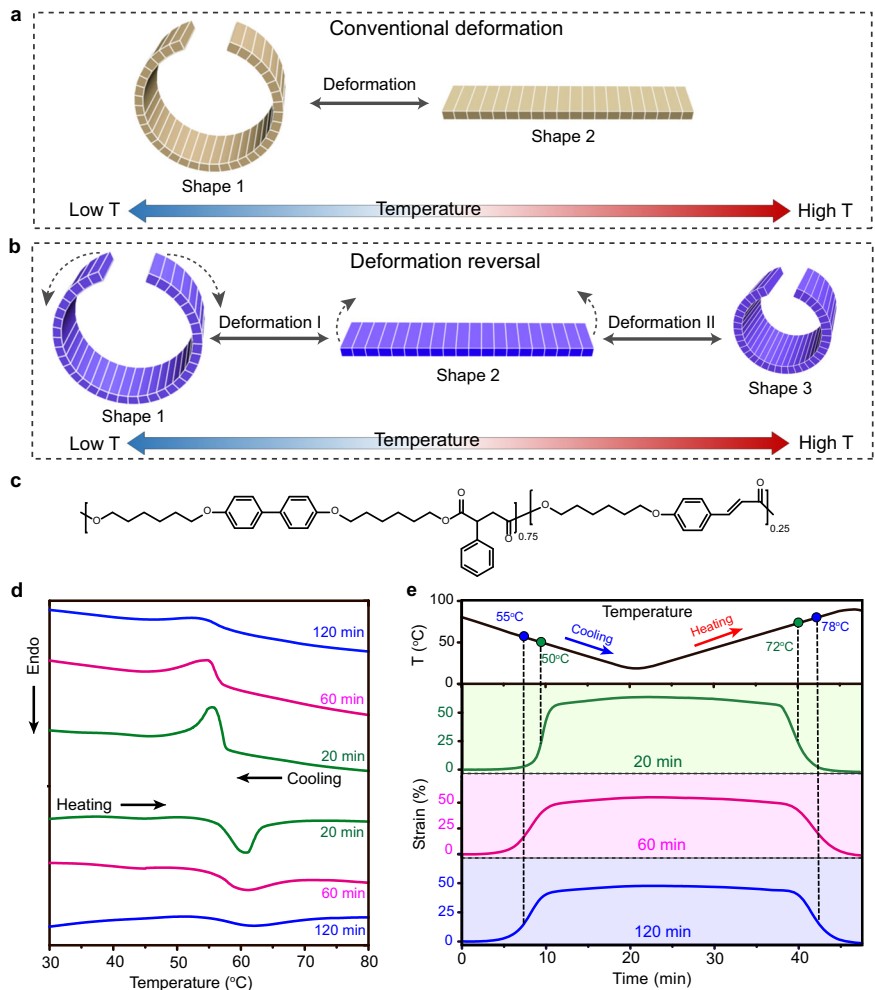

**Fig. 1 Concept of deformation-reversal behavior and crosslinking-determined mechanical response of LCN actuators. a** Conventional shape change with only one deformation direction during one heating (or cooling) process. **b** Deformation reversal in LCN actuators: by only heating (or cooling), the actuator possesses two deformation stages with contrary shape evolution directions, generating three characteristic shapes with a deformation reversal. **c** Chemical structure of the polymer used for the LCN actuators. **d** Differential scanning calorimetry (DSC) curves of the LCN strips with each side photocrosslinked for 20 min (green), 60 min (magenta), and 120 min (blue), recorded in the second heating scan (bottom) and first cooling scan (top). **e** Thermally induced elongation and contraction under controlled cooling and heating (3 °C min⁻¹), respectively, for LCN strips with each side photocrosslinked for 20 min (green), 60 min (magenta), and 120 min (blue), the strain being measured with the strip length in the isotropic state as the reference length. Note: Unless otherwise stated, the dashed arrow in a drawn cartoon shape indicates the deformation direction leading to the adjacent shape on the right.

heating into the isotropic state. However, the changing crosslinking gives rise to notably different actuation behavior. The LCN crosslinked for 20 min on each side shows lower mid-point actuation temperatures on cooling (50 °C) and on heating (72 °C), determined as the temperature at which 50% of the reversible strain occurs, while the LCN crosslinked for 120 min on each side shows higher mid-point actuation temperatures (55 °C on cooling and 78 °C on heating). In addition to the actuation temperature range difference, less crosslinked LCN shows larger reversible deformation degree[47,48], as can be seen from the plateau strain in Fig. 1e, being 64%, 55%, and 48% for LCNs crosslinked for 20, 60, and 120 min on both sides, respectively (the measurement is detailed in Supplementary Methods). As shown below, such differences in actuation temperature and deformation degree enable the desynchronized actuation leading to the deformation-reversal behavior of LCN actuators.

**Deformation-reversal behavior**. Firstly, the deformation-reversal actuators with bending-based motions were fabricated by

stretching the LCNs to 400% (draw ratio λ) followed by crosslinking both sides of the film for different times, thus crosslinking degrees. The prepared actuators are coded as LCN₁(X,Y) (X and Y represent the crosslinking time of side A and side B, respectively, and the subscript 1 denotes the first series of samples used for varying the crosslinking dissymmetry of the two sides while keeping a fixed λ at 400%). Figure 2a and Supplementary Movie 1 show a typical deformation-reversal process of LCN₁(20,120) during cooling from 75 to 40 °C, which can be explained by the crosslinking-dependent actuation behaviors shown in Fig. 1e. In isotropic phase, the film exhibits a bent shape toward side A because the lightly crosslinked side A contracts more than the highly crosslinked side B. On cooling from 75 to 57 °C, the highly crosslinked side B elongates while the lightly crosslinked side A shows no obvious deformation, which causes the actuator to curl to side A and generates a coiled structure (deformation I). When the temperature drops below 57 °C, the bending direction of the sample reverses to side B as the extension of the side A proceeds with a higher strain (deformation II). Hence, in the cooling

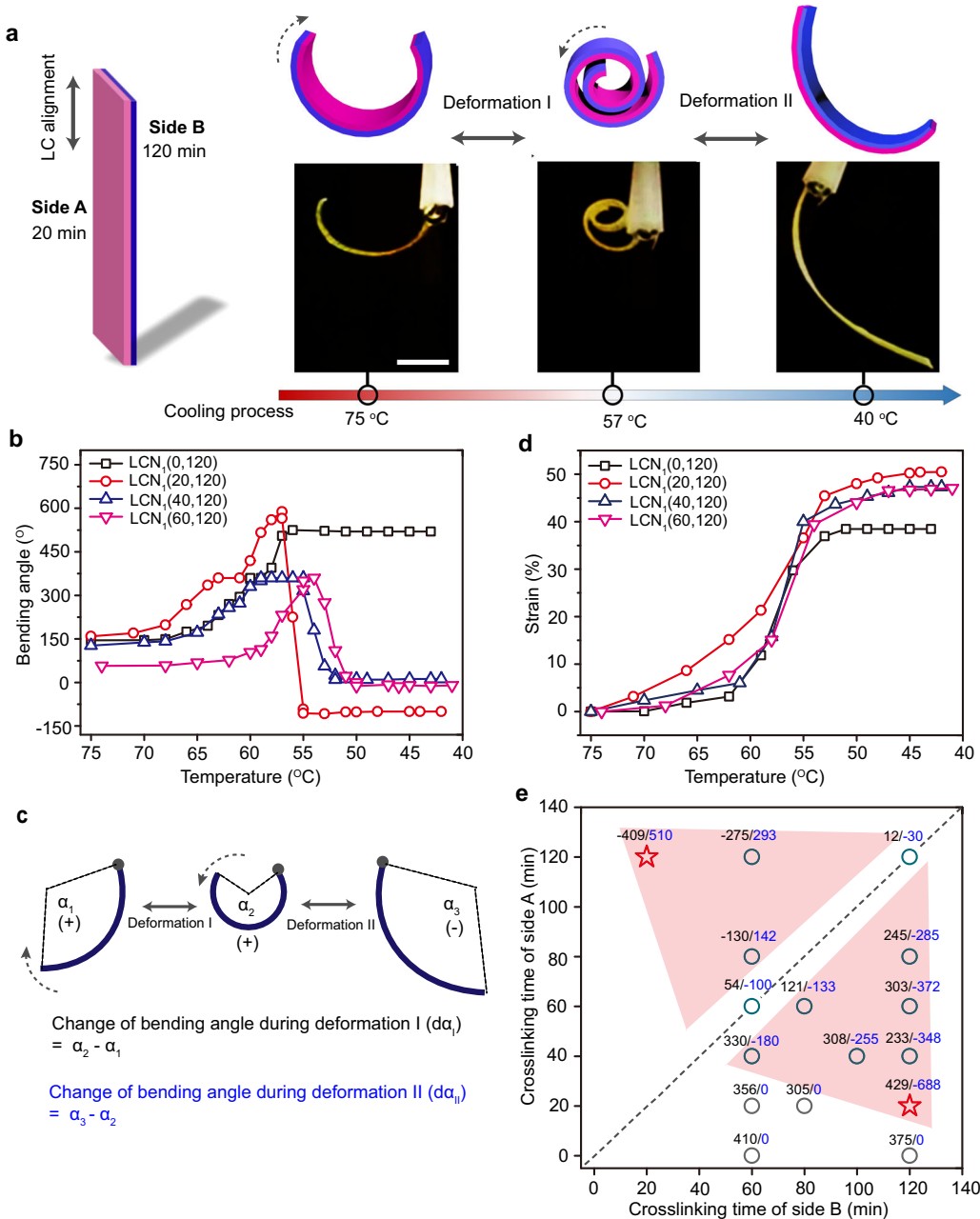

**Fig. 2 Deformation reversal based on asymmetrical crosslinking. a** Schematics and photographs (scale bars: 2 mm) of a planar aligned LCN$_1$(20,120) actuator displaying deformation reversal during cooling process (see also Supplementary Movie 1). **b** Bending angle of LCN$_1$(X,Y) as a function of temperature recorded during cooling. **c** Schematics showing how to define and determine $\alpha_1$, $\alpha_2$, $\alpha_3$, d$\alpha_I$, and d$\alpha_{II}$. Note: Bending toward side A is defined as positive while bending toward side B is defined as negative. **d** Plots of strain (measured with the strip length in the isotropic state as the reference) versus temperature of LCN$_1$(X,Y) actuators. **e** A deformation-reversal diagram showing the crosslinking-determined bidirectional bending, as revealed by d$\alpha_I$ (black) and d$\alpha_{II}$ (blue), for LCN$_1$(X,Y) actuators during one cooling run. The colored regions show the crosslinking conditions required for realizing prominent deformation-reversal behavior.

process, the LCN$_1$(20,120) undergoes both deformation I and II stages having opposite bending directions and realizes the transformation among three different shapes. On subsequent heating, the actuator executes the opposite deformation-reversal steps (Supplementary Fig. 2 and Movie 1).

To investigate the factors that dictate the deformation-reversal behavior, we prepared different samples with fixed crosslinking time of side B (120 min) and variable crosslinking time of side A (0, 20, 40, and 60 min). The change in bending angle of the LCNs as a function of temperature is shown in Fig. 2b. Since the bending angle is related to the geometry of the specimen[49], for a

bending angle comparison to be meaningful in characterizing the deformation-reversal behavior, we kept similar length and thickness for the different LCN actuators (see Supplementary Table 1). As expected, since the uncrosslinked side shows no reversible deformation, LCN$_1$(0,120) can only bend unidirectionally during cooling. In contrast, the three samples crosslinked on both sides all exhibit deformation-reversal behavior, i.e., bidirectional bending. As the crosslinking-time difference increases, the deformation-reversal behavior becomes more pronounced with larger changes in bending angle during deformation I ( $|$ d$\alpha_I|$ ) and deformation II ( $|$ d$\alpha_{II}|$ ) (Fig. 2c). Furthermore,

accompanying the bending-reversed bending process on cooling, all samples crosslinked on both sides extended about 50% of their length in isotropic phase (Fig. 2d). Further expanding the combinations of crosslinking times for side A and side B of the LCN (see Supplementary Fig. 3–5), a graph showing the influence of the crosslinking time on the actuation behavior was obtained, on which the largest bending angle changes in the two directions, $d\alpha_I$ and $d\alpha_{II}$, are listed for each actuator (Fig. 2e). As can be seen, to obtain larger $|d\alpha_I|$ and $|d\alpha_{II}|$ and more significant deformation-reversal behavior (see the colored region in Fig. 2e), the crosslinking times of the two sides should be adjusted to meet two criteria: (1) large difference in crosslinking time, and (2) effective actuation via sufficient crosslinking on both sides of the film. Therefore, the optimal crosslinking condition shown in the map is 20 min/120 min on the film's two sides. The area near the diagonal dashed line indicates insufficient crosslinking-time difference; the area close to while at the far end of either coordinate axis (meaning one side highly crosslinked while the other barely crosslinked) cannot meet the second criterion; the area near the origin of the coordinate could fail to satisfy both criteria. A remark must be made here. The shape change of LCN actuators is often characterized using bending/unbending, simply because this is the most visible deformation. The actual shape change is a continuous 3D deformation of the soft LCN material matrix (i.e., change in geometry, length, thickness, and width).

As the deformation-reversal capability is structurally inscribed in the LCNs via tuning the crosslinking times, the resulting actuator is stable and displays the deformation reversal under repeated heating/cooling cycles (20 cycles monitored, Supplementary Fig. 6). Although the recorded $|d\alpha_I|$ and $|d\alpha_{II}|$ decreased slightly, which could be attributed to some unrestored order during actuation under the used conditions, the initial performance can be recovered by simply setting the actuator in air at room temperature for a couple of hours. Some other factors, such as film thickness, stretching temperature and draw ratio ($\lambda$, larger than 400%), have been investigated to further verify the decisive role of asymmetrical crosslinking in imparting the LCN actuator with the deformation-reversal capability. When fixing the crosslinking times at 20 and 120 min for the two sides of the strip, respectively, the deformation-reversal behavior was observed for all samples investigated (including two LCN actuators prepared from linear polymer films with original thicknesses of 0.20 mm and 0.48 mm), although the performance of the actuator varies (Supplementary Fig. 7). Moreover, as mentioned above, the LCN used possesses initially a smectic phase (Supplementary Fig. 8). For comparison, we synthesized an LCN with only a nematic phase (Supplementary Fig. 9, synthesis procedures are detailed in Supplementary Methods). The nematic LCN actuator also exhibits deformation-reversal behavior (see Supplementary Fig. 10), confirming that the desynchronized actuation can be applied to different LCNs.

When a monolithic LCN film is subjected to asymmetrical photocrosslinking, its two sides not only differ in average crosslinking density but are likely to have different crosslinking depths along the thickness direction with a density gradient. Through the gel fraction measurement (Supplementary Fig. 11 and 12), the crosslinking depth was estimated to be between 0.010 and 0.071 mm with increasing photocrosslinking time from 20 to 120 min under the UV light intensity used (320 nm, 180 mW cm$^{-2}$). However, it is hard to determine precisely the crosslinking depth and the crosslinking gradient. For a given dimension of the LCN strip, the deformation-reversal behavior is governed by the competition between the lightly and highly crosslinked layers likely sandwiching an uncrosslinked layer in between. This competition could be a complex interplay of the strain difference, geometry change, mechanical properties (Supplementary Fig. 13) and internal stress, which are all temperature-dependent, leading to a range of bending angle changes as shown in Fig. 2e. On the basis of this principle, it can be expected that similar deformation-reversal behavior could be obtained for LCNs with different thicknesses, but the necessary photocrosslinking times vary.

**Enhancing deformation-reversal capability**. As demonstrated above, by tuning the crosslinking times for both sides of a uniaxially stretched LCN, we can control the direction and extent of the deformation-reversal behavior, corresponding to the sign and absolute value of $d\alpha$, respectively. However, the obtained shape change conforms to either $\alpha_1 > 0$, $d\alpha_I > 0$, $d\alpha_{II} < 0$ (if X < Y), or $\alpha_1 < 0$, $d\alpha_I < 0$, $d\alpha_{II} > 0$ (if X > Y). This exposes the limitation of adjusting the deformation-reversal behavior only by changing the crosslinking times. This is because crosslinking not only determines the signs of $d\alpha_I$ and $d\alpha_{II}$ (the shape change direction), but also affects the signs of $\alpha_1$ (the shape in isotropic phase). In this regard, we further enriched the attainable deformation-reversal behaviors of our LCN films by not only generating crosslinking difference, but also creating draw-ratio difference on both sides of the LCNs. Typically, a strip was stretched to a draw ratio and one side was exposed to UV light for photocrosslinking; then the strip was stretched again to a larger draw ratio and the other side was photocrosslinked. In other words, the draw ratio on one side indicates the strain at which that side is photocrosslinked, though the final strain on both side is the same for the LCN strip. As shown in Fig. 3a, the crosslinking times of side A and B for all samples were fixed at 20 min and 120 min, but for the draw ratio, the $\lambda$ of side B was fixed as 400% for all samples, while that on side A varied from 0% to 1000%. The obtained actuators are named as LCN$_2$(M,N) (M and N represent the $\lambda$ of side A and B respectively, and the subscript 2 denotes this second series of actuators). As seen from the photographs in Fig. 3a, we achieved different types of deformation-reversal behaviors. For example, LCN$_2$(700%,400%) can complete a $-360°$ to $360°$ to $-360°$ deformation mode during one cooling run; LCN$_2$(1000%,400%) undergoes ring to flat to ring two-stage deformation. These LCN actuators all show deformation where $\alpha_1 < 0$, $d\alpha_I > 0$, $d\alpha_{II} < 0$. It is easy to notice that the $\alpha_1$ of LCNs with $\lambda$ of side A greater than 400% are shifted to negative values, indicating that, as $\lambda$ increases, the shape in isotropic phase can be changed from rolling toward side A to curling toward side B (Fig. 3b). This is understandable because a higher $\lambda$ results in a longer LCN actuator. Concurrently, high bending angle changes (large $|d\alpha_I|$ and $|d\alpha_{II}|$) were achieved. Among them, the $|d\alpha_I|$ and $|d\alpha_{II}|$ of LCN$_2$(540%,400%) reach up to 770° and 735°, respectively (Supplementary Fig. 14a). In comparison, the samples with low $\lambda$ of side A show no or only slight deformation-reversal behavior due to the lack of good mesogen alignment which leads to ineffective/insufficient actuation on side A and an unmet second criterion (see Fig. 3b and Supplementary Fig. 14). Therefore, the above results demonstrate effective enhancement and tuning of the deformation-reversal behavior through adjusting the $\lambda$ of the two sides prior to crosslinking of a monolithic LCN film. It should be noted that the slight differences in length and thickness of these samples (see Supplementary Table 1) can cause a certain variation in the observed bending angles, but the overall trend is not affected. A reasonable assumption is that the deformation-reversal behavior can be regulated on-demand by fine-tuning these two processing factors (crosslinking time and $\lambda$), generating a large variety of shape change modes. More interestingly, when encoding the asymmetrical crosslinking and $\lambda$ on the two halves of the LCN along its width direction instead of its two sides along the thickness, controllable in-plane bending and bending reversal can also be readily realized (see Supplementary Fig. 15).

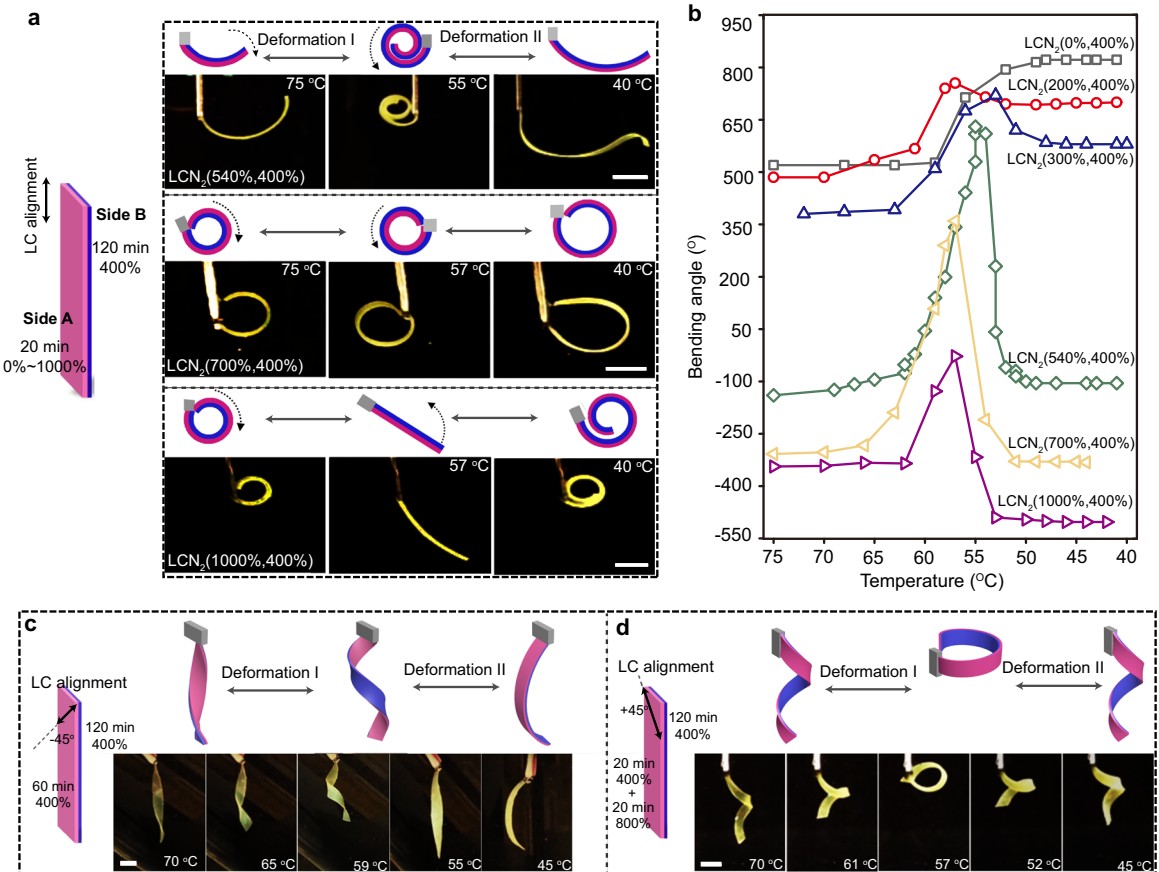

**Fig. 3 Tuning of deformation-reversal behavior. a** Schematics and photographs of planar aligned LCN$_2$(540%,400%), LCN$_2$(700%,400%) and LCN$_2$(1000%,400%) actuators displaying diverse deformation-reversal behaviors during cooling (see also Supplementary Movie 2). **b** Bending angle changes of LCN$_2$(M,N) versus temperature during cooling process. **c** Schematics and photographs of an asymmetrically crosslinked LCN actuator with the LC director at a −45° angle with respect to the long axis of the strip, displaying twisting-untwisting two-stage deformation during cooling (see also Supplementary Movie 3). **d** Schematics and photographs of an asymmetrically stretched and crosslinked LCN actuator with the LC director at a +45° angle with respect to the long axis of the strip, displaying untwisting-twisting two-stage deformation during cooling (see also Supplementary Movie 4). Scale bars: 2 mm.

Not limited to simple bending-based deformation-reversal behavior, we also achieved twisting-based deformation reversal using the LCN strip with tilted LC orientation relative to its long axis. As shown in Fig. 3c, the LCN$_1$(60,120) having its mesogens aligned −45° with respect to the principal axis of the strip shows a slightly twisted form in isotropic phase, which then winds up and unwinds in the following cooling process. Furthermore, Fig. 3d presents a contrary deformation-reversal behavior. The film was prepared by firstly crosslinking a LCN strip (λ = 400%, +45° mesogen alignment) for 20 and 120 min on side A and B, respectively, then stretching the film to double its length and further crosslinking the side A for 20 min. The resulting LCN displays a helical shape in isotropic phase, with the highly crosslinked side as the inner of the helix. Upon cooling, the helical LCN first unwinds and then winds up, showing opposite deformation trend compared with that shown in Fig. 3c.

**Photomobility of deformation-reversal actuators.** The central interest of the deformation reversal resides in the fact that the number of bidirectional shape switch can be doubled under one stimulation on/off cycle. Since repeated shape switch under repeated on/off stimulation is the basis of moving LCN soft robots, this feature can be explored. Therefore, on the basis of the diverse forms of the attainable deformation-reversal behaviors, we further went on

to demonstrate that these thermal responsive shape changes can also be controlled by light. To this end, a thin layer of polydopamine (PDA) (ca. 130-nm thick) was coated on the crosslinked LCNs to serve as photothermal agents (see Supplementary Figs. 16 and 17 for more characterizations). The PDA-coated LCN film showed unchanged thermal-induced actuation behavior (see Supplementary Fig. 18), while possessing similar but more rapid light-driven deformation-reversal behaviors based on the photothermal effect, regardless of which side is exposed to the light source (see Supplementary Fig. 19–20). In contrast to the conventional light-driven actuation, our LCN actuators can bend and unbend (or unbend and bend) twice in one light-on/off cycle (see Supplementary Fig. 21 and Supplementary Movie 5). As light allows remote and convenient control of the fast and reversible deformation, light-guided locomotion based on deformation-reversal behavior has been explored. Shown in Fig. 4a, b is a miniature walker of 3.5 mm in length, which is made of PDA-coated LCN$_1$(20,120). The sample is initially flat with the highly crosslinked side facing upward. After exposure to light (30.5 mW mm$^{-2}$) for ~2 s, the film arches up with the right side as the stationary point; further exposure for ~1 s, the film flattens down with the left side as the stationary point. As such, the film already takes a step forward with the light spot on it. Ceasing the light, the film cools down and reaches up with the right side as the stationary point, followed by flattening down as it cools further. In this way, the film can implement two steps forward in one

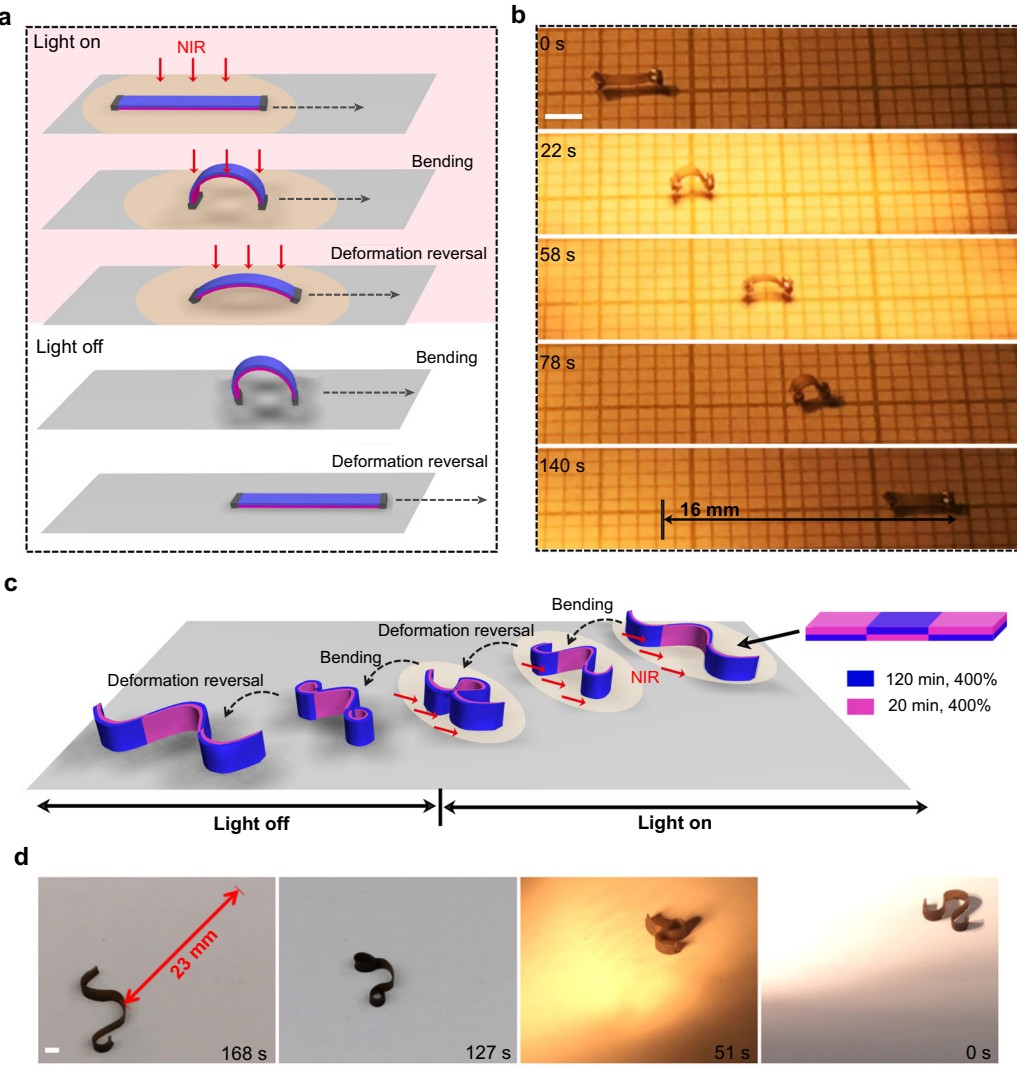

**Fig. 4 Light-driven walkers with doubled step numbers.** Schematics (**a**) and photographs (**b**) of polydopamine-coated LCN$_1$(20,120) microwalker, placed with one side on the substrate, which can advance two steps through arching up/flattening down in a single on/off irradiation cycle (see also Supplementary Movie 6, scale bar: 2 mm). Schematics (**c**) and photographs (**d**) of a patterned, wave-shaped walker with one edge on the substrate, which can move based on deformation reversal-induced four-stage deformation in one light-on/off cycle (see also Supplementary Movie 7, scale bar: 2 mm).

light-on/off cycle. Due to the doubled number of steps, the micro-walker can move at a speed of ~2 body length min$^{-1}$. Combining photopatterning with asymmetrical crosslinking for deformation-reversal behavior, we further achieved a wave-shaped LCN walker that can move in-plane with one edge on the substrate without "body" lifting in air (Fig. 4c, d). The LCN, prepared to have three sections with alternating highly crosslinked sides (120 min, 400%) and lightly crosslinked sides (20 min, 400%), can first fold into a compact accordion shape that then expands into a loose wave under continuous light irradiation; upon retrieving the light, the loose wave tights up into an accordion shape and further expands into a large wave shape, thus completing one shape change cycle (see Supplementary Movie 7 for the shape change process). When this actuator is obliquely exposed to light-on/off cycles from one side as shown in Fig. 4c (light of 16.8 mW mm$^{-2}$, shining from ~25° relative to the substrate surface), it can walk with the two end sections performing deformation reversal to adjust the friction during motion, while its middle part, being self-shadowed after curling up, displays only unidirectional deformation (speed: 8 mm min$^{-1}$, see Supplementary Fig. 22 and Supplementary movie 7). It should be emphasized that, unlike laser scanning process where only part of the actuator

is activated at any point in time, the light is evenly applied to the actuator in the present case. Therefore, there is no need to precisely position the light spot on the micro-scale device, which is beneficial to light-fueled locomotion[50].

**Multimodal light-driven locomotion.** The deformation-reversal behavior also imparts light-fueled LCN soft robot with high versatility in motion. This is because the entire shape change process with doubled number of shape switches can be divided into many stages by simply adjusting the time period of light-on and light-off, corresponding to different actuation temperature oscillation profiles. As a result, different movement fashions, driven by side-on out-of-plane bending/unbending and/or edge-on in-plane bending/unbending, can readily be achieved using the same LCN actuator under light exposure of constant intensity. To simplify the discussion, these different locomotion patterns involving different types of actuator deformation will be referred to as different modes. As a proof of concept, we were able to obtain five types of motion patterns using a single actuator prepared as illustrated in Fig. 5a: one side of the strip was crosslinked for 120 min at λ = 400%, it was then further stretched to λ = 540% and had the two halves of the

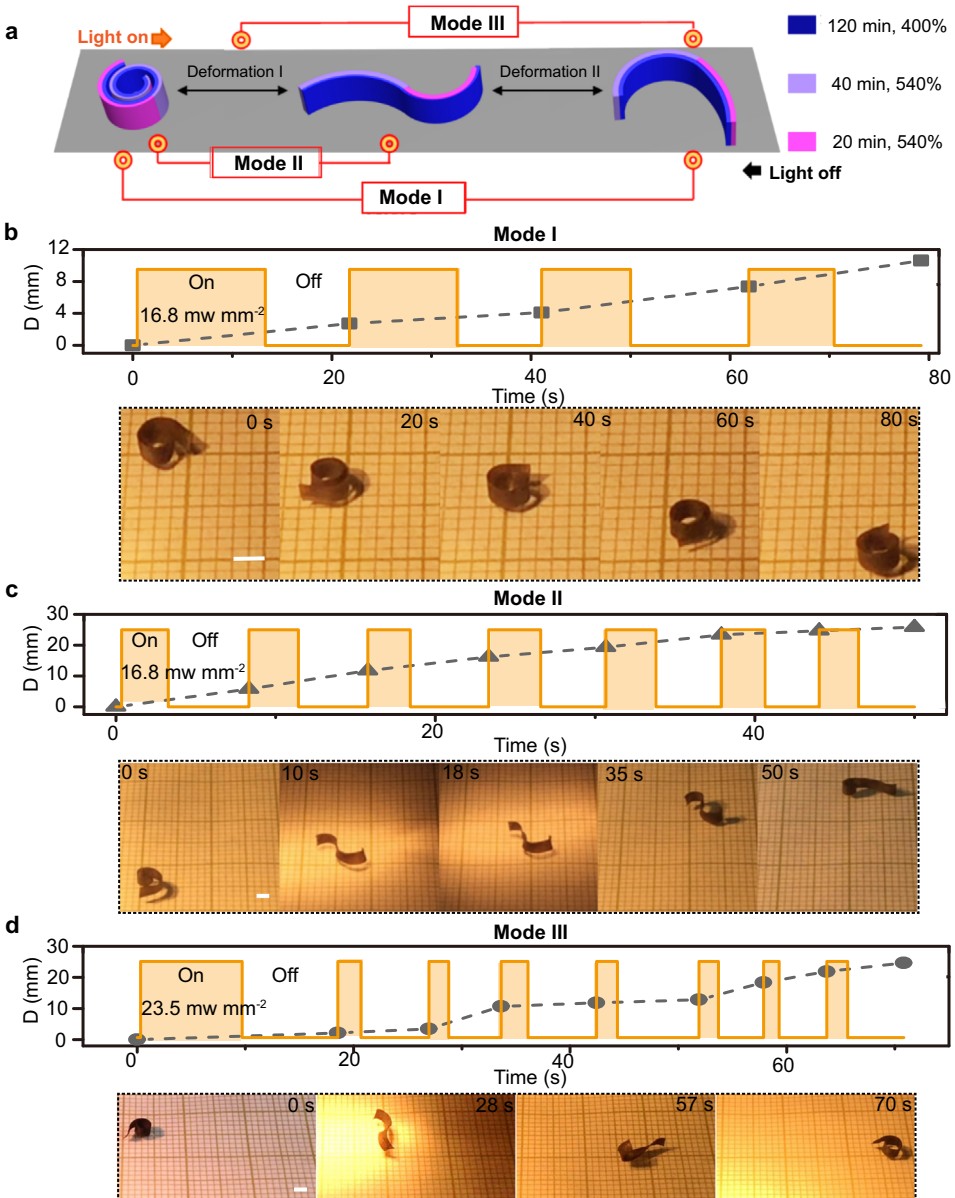

**Fig. 5 Multimodal locomotion of a single deformation-reversal LCN actuator. a** Schematics of an actuator prepared to exhibit reversible switching among three different shapes under one light-on/off cycle. **b–d** Displacement (D) versus time and light irradiation pattern (top) as well as photographs (bottom) showing different locomotion modes for the same actuator: **b** moving through full two-stage shape change leading to body rotation (Mode I); **c** walking based on unidirectional stage I deformation (Mode II); and **d** moving through part of two-stage deformation and imbalanced turning over (Mode III) (see also Supplementary Movie 8). Scale bars: 2 mm.

other side crosslinked for 20 and 40 min, respectively. When exposed to light, this actuator can evolve from a coil shape to an S shape and then to a C shape, while when light is off, it transforms from the C shape to the S shape and then to the coil shape (Supplementary Fig. 23a). The reversible shape change can be made to occur between different stages of the complex shape morphing process to realize different locomotion modes. For the sake of clarity, only three modes are shown in Fig. 5 and Supplementary Movie 8 (2 others in Supplementary Fig. 24 and 25). In Mode I (Fig. 5b), the actuator is subjected to repeated entire shape change process (coil-S-C) under sufficiently long light-on and light-off times (light of 16.8 mW mm$^{-2}$), which results in its motion through body rotation (visible from the changing state at the end of each light-on/off cycle, at a speed of 8 mm min$^{-1}$). In mode II (Fig. 5c), the light-on and light-off times are shortened to enable the

reversible shape switch only between coil and the S shape (light of 16.8 mW mm$^{-2}$). Under this condition, the actuator moves through the unidirectional uncurling (light-on) and curling (light-off) movements, at a faster speed of 31 mm min$^{-1}$. In Mode III (Fig. 5d), the light irradiation condition is set to generate the back-and-forth switch between the shape C (deformation II) and an intermediate shape during deformation I (after obtaining shape C, light-off time is controlled to prevent the complete shape restoration during cooling). In this case, the imbalance in shape evolution could be enhanced by the higher light intensity 23.5 mW mm$^{-2}$ used and the incomplete shape change, which causes the actuator to advance by flipping along its long axis, with an intermediate loco-motion speed (21 mm min$^{-1}$). Moreover, when the actuator lying on its edge flips under light irradiation and has its planar side parallel to the surface, it can also rotate (Mode IV shown in

Supplementary Fig. 24) or walk (Mode V shown in Supplementary Fig. 25) when the entire or partial deformation process is activated, respectively (see also Supplementary Movie 9).

## Discussion

The key to imparting a monolithic LCN actuator with the deformation-reversal capability is an appropriate desynchronization of the two sides in terms of actuating temperature (starting temperature and temperature range) and actuating strain (degree of reversible deformation). We show that this can readily be obtained by crosslinking the two sides to different crosslinking densities (varying photocrosslinking time) and/or by crosslinking the two sides at different mechanical stretching (elongation strain). Once prepared, the desynchronized actuation and the resultant deformation-reversal behavior are structurally inscribed in the LCN actuator, and it requires a uniform heating and cooling of the actuator over the thermal order-disorder phase transition to activate the actuation. This means that, in principle, the deformation reversal and its enabled intricate shape switch, twice bidirectional shape shifts on one stimulation on/off cycle as well as multimodal locomotion can be achieved by either direct heating/cooling, or light-on/off, or electric field on/off or magnetic field on/off, all this through a stimulation-induced thermal effect. A case worth discussing and comparing is the widely reported bilayer or multilayer actuators that naturally have different actuator sides. In addition to the use of two or more layers of often different polymers, generally, one side layer is a passive layer and used to adjust the mechanical response and/or to facilitate the shape processing[33], and they show no deformation reversal. Combining two active layers may achieve deformation-reversal behavior, but there are some foreseeable problems: (1) Extra effort is needed for bonding the two active layers. (2) Even if two active layers are combined in one actuator, the two layers still need to be specifically regulated to satisfy the two criteria mentioned above to exhibit deformation reversal. Otherwise, the shape change process may be dominated by one of the active layers. (3) Well-tuning of the deformation-reversal behavior may still need to rely on adjusting the processing factors of the different layers, while tuning by changing the polymer structure or composition is more complicated. Nevertheless, in light of the present study, this is possible and remains to be realized in the future.

In conclusion, we report an atypical monolithic LCN actuator that can complete the shape switch (over order-disorder phase transition) twice in one stimulation on/off cycle, instead of once for known LCN actuators. Basically, in one reversible deformation cycle, the LCN actuator can possess four deformation stages with the deformation direction reversed three times, involving the conversion among three shape states. The required deformation direction reversal is obtained by applying the desynchronized actuation strategy, meaning that the two sides of an LCN strip are purposely prepared to start deforming at different temperatures and with different strains. We show that this property can be obtained through asymmetrical crosslinking of the two sides or through an asymmetrical stretching/crosslinking process (a first stretching and one side crosslinking, followed by a second stretching and crosslinking of the other side). Introducing asymmetrical draw ratio on top of the asymmetrical crosslinking to the two sides of an LCN can further enhance the tunability of the deformation parameters and enrich the attainable deformation-reversal behaviors. Combining with the tilted mesogen alignment and photopatterning techniques, the deformation-reversal feature can be imparted to more sophisticated geometries, like helix and wave shapes. More importantly, this intriguing actuation characteristic can be exploited for light-guided locomotion of LCN soft robots. By simply adjusting the light-on and light-off times, which determines the deformation stages, the same LCN actuator can exhibit multiple modes of locomotion with varying speed. This study starts from the basis of the actuation mechanism of LCN actuators, but adopts another perspective to realize a distinct deformation feature that could potentially be extended to various LCN systems and beyond, which can greatly expand the soft robotics development in terms of fabrication, control, and operation.

## Methods

**General considerations**. All used reagents, polymer synthesis and characterizations are described in Supplementary Methods. The LC monomer 4,4′-bis(6-hydroxyhexyloxy)biphenyl (BHHBP) and the crosslinker 4-(6-Hydroxy-hexyloxy) cinnamic acid (6HCA) were synthesized according to the literature procedures; the synthetic protocol and NMR spectra of the liquid crystalline polymer (LCP) used in this study are included in Supplementary Figs. 26–27 and Supplementary Methods.

**Preparation of LCN₁(X,Y) actuators**. Using a homemade device, the compression-molded polydomain LCP film (thickness 0.35 mm) was stretched uniaxially at 58 °C (in LC phase) to a draw ratio ($\lambda$) of 400%. Then the side A and side B of the film with uniaxial orientation of mesogens were exposed to UV light (320 nm, 180 mW cm$^{-2}$) at 40 °C (above Tg) for X and Y min, respectively, for photocrosslinking. It should be noted that after the compression molding, the two sides of the film are marked as A and B, respectively. This is to standardize the operation details in the subsequent preparation process of the material as much as possible and reduce the error.

**Preparation of LCN₂(M,N) actuators**. LCN₂(0%,400%) was prepared by firstly crosslinking the side A of the unstretched polydomain LCP film for 20 min, then stretching the film to $\lambda$ = 400% before crosslinking the side B for 120 min. Other LCN₂(M,N) actuators were obtained through the two-step elongation and crosslinking procedures. By stretching the LCP film to $\lambda$ = 200% or 300% and crosslinking the side A for 20 min, restretching the film to $\lambda$ = 400% and irradiating side B for 120 min, LCN₂(200%,400%) and LCN₂(300%,400%) were fabricated. To prepare LCN₂(540%,400%), LCN₂(700%,400%), and LCN₂(1000%,400%), the LCP film was stretched to $\lambda$ = 400% and side B of the film was crosslinked for 120 min, followed by restretching the film to $\lambda$ = 540%, 700%, or 1000% and irradiating the side A for 20 min.

**Preparation of the light-fueled LCN walkers**. The LCN actuators coated with polydopamine (PDA) were obtained according to a literature method[51]. Dopamine hydrochloride (0.2 g, 1.05 mmol) and Tris base (0.1 g, 0.83 mmol) were dissolved in 100 mL of water. Afterward, the LCN actuators were immersed into the solution and stirred for 72 h (24 h for actuator shown in Fig. 4b). The resulting PDA-coated LCN films were washed with deionized water three times and air-dried. The wave-shaped actuator shown in Fig. 5b was prepared by irradiating the patterned LCP stretched to $\lambda$ = 400%.

## Data availability

Source data are provided with this paper.

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

## Acknowledgements

Y.Z. acknowledges financial support from the Natural Sciences and Engineering Research Council of Canada (NSERC) and le Fonds de recherche du Quebec: Nature et technologies (FRQNT). Y.X., Z.J. and J.H. thank FRQNT and China Scholarship Council (CSC) for awarding them a scholarship.

## Author contributions

Y.Z. conceived the idea. Y.X., Z.J., J.H. and Y.Z designed the research and analyzed the results. Y.X. and Z.J. performed the experiments. All authors contributed in writing the manuscript.

## Competing interests

The authors declare no competing interests.
