## [Peer Review File · Nature Communications]

REVIEWER COMMENTS

Reviewer #1 (Remarks to the Author):

The authors report a new type of liquid crystal network (LCN) actuator that deforms twice i.e. having three shapes upon applying one stimulus. This so-called deformation reversal has not been reported before and is interesting both from a fundamental and application point of view. Up to date only actuators have been described that have two shapes upon applying a single stimulus. The new actuators for example can be switched between a ring shape, a flat strip and a reverse ring shape of different size by changing only the temperature. The deformation reversal is programmed through asymmetrical crosslinking and/or asymmetrical stretching of the two sides of the LCN actuator. Furthermore, by using a photothermal coating, light can also be used to induce the shape changes. These novel findings lead to new unique possibilities in developing actuators and soft robot devices. I recommend publication of this well written manuscript after addressing the following issues.

- 1) It is not clear why an increasing crosslinking leads to widening of the LC-isotropic phase transition temperature range while the peak position does not change. The authors should investigate this behavior in more detail as this is the base of the desynchronized actuation.
- 2) The authors write that the performance of the actuators does not depend on thickness of the polymer film. However, the deformation should depend on the thickness of the low and high crosslinked layers. Do the authors have an explanation? Furthermore, what is the thickness of the two layers and is there also a region in the middle of the sample where the film is not crosslinked.
- 3) The reversibility and fatigue of the actuators is not clear. In most cases only cooling data is presented (for example, Figure 2, 3). The actuation behavior upon heating should also be given. How many times can the actuation be repeated?
- 4) The characterization of the polydopamine actuators is poor. In the experimental only the preparation procedure is given.

Reviewer #2 (Remarks to the Author):

In this manuscript entitled “Desynchronized Liquid Crystalline Network Actuators with Deformation Reversal Capability”, the authors provided a desynchronized actuation strategy by endowing the two sides of a monolithic LCN with different crosslinking densities or by crosslinking the two sides at different elongation strain, to prepare a novel type of LCN, which could reversibly deform among three different pre-set shapes on heating or under light irradiation. The most significant contribution of this work is that (1) it realizes the multi-stage shape changes of a monolithic LCN film in one stimulation on/off cycle; (2) the asymmetrical crosslinking methods can be potentially extended to various actuator systems to design and manufacture soft robots which could complete complex motions under the control of a simplex external stimulus. The paper is well organized and written. The conclusions are well supported by the convincing data and characterization. Overall, I believe the work will attract interest of a broad readership and will have far reaching impact in LCN research field. Therefore, I strongly recommend publication of this manuscript after the following minor revisions (listed below) are addressed:

Questions:

1. Scale bars should be added in the photo images of the actuators in Figure 4 and Figure 5.
2. The penetration of UV light is relatively poor, which might cause a gradient of crosslinking density along the thickness direction of the LCN film. What is the relationship between the thickness of the LCN films and their responsive behaviors?
3. Both the phrases “liquid crystalline networks” and “liquid crystal networks” are used in this manuscript. Although both phrases are used by researchers in this field, it is better for the authors to unify them in order not to cause confusion in more extensive audience.
4. What if we prepared a fiber using the strategy of this work? Could the multi-stage deformations be realized?

Reviewer #3 (Remarks to the Author):

The paper “Desynchronized Liquid Crystalline Network Actuators with Deformation Reversal Capability” by Yue Zhao, et al, reported a fabrication of asymmetric liquid crystal network actuator that can exhibit bending and reversed bending upon heat (or light) stimulus. The deformation reversal is achieved by tuning the phase transition temperatures between two sides, rendering a so called “desynchronized” process. Detailed experiments (controlling the polymerization time and stretching) are applied to control such desynchronized actuation behaviors. Applications in multi-modal walking robot are demonstrated. The story is well written, providing certain advances to the field. Hopefully the following comments could help to further improve the quality.

The concept in general.

I am a bit skeptical about the novelty level raised by the authors. I think it is inappropriate to make a strong point at the beginning, by saying, “peculiar and fascinating actuation behavior”, “has not been conceptualized and explored before”. i) If an actuator is constructed with two sides of different materials that contract at different temperatures, it should firstly bend and then unbend. This phenomenon is quite obvious, at least to me. ii) There are some similar demonstrations reported before, as the authors also mentioned at the very last section of the manuscript (line 350).

So, my suggestions are,

- i) Bring Ref. 44-46, the studies about bend-unbent events in LCN, to the introduction for comparison. The authors can still claim they are “the first one”, but, to highlight such deformation reversal mechanism and use it in soft robotic purposes. The solid scientific pieces in this manuscript are the fact that, the authors have developed comprehensive experimental procedure by changing the polymerization time and stretching ratio, to obtain a well-control of such deformation reversal.
- ii) What really distinguish the deformation reversal reported in this work from those in Ref.44-46 is that, the bending (at certain temperature) is obtained/measured at thermal equilibrium. Many thermal or photo responsive LCN actuators, may exhibit bending-to-unbending deformation upon one stimulus, only because they are moving toward the thermal equilibrium (in the case of change of thermal gradient, or say, when the heat is going from one surface to another), or photo stationary state (in the case of cis azo population across the thickness). Those bending-unbending deformation are just intermediate states during the time span of response evolution. I believe such difference is significant, however, not easy to explain in few words.
- iii) Line 43. the authors mentioned in the introduction, the typical way of shape change is from

shape1 to shape2. I understand and also agree with the authors on this. But, readers may confuse with the reconfigurability concept which has been widely reported in LCNs, since they can produce different shapes upon the same stimulus.

Now, the introduction is very short, I strongly recommend the authors to twist the points listed above, and reshape the introduction. After that, the novelty should be more clear, and easier for readers to appreciate.

Main points:

1. All the deformations are characterized by measurement of the bending angle. However, the bending angle of a long strip is so sensitive to the thickness and length. All the samples are prepared with 350 microns thick, BUT, after the stretching, thickness and length both change. This actually brings some hurdles and complexity in comparing data (Fig.3b). Although these parameters won't bring a "decisive" effect as the authors mentioned, they do affect the bending angle characterization, especially in the experiments for a well-control of such behaviors (Fig.2e). I would suggest the author to provide thickness, length data of each actuator, and more discussion on this. It is unrealistic to repeat all experiments by using samples with the same thickness. So, this comment doesn't require extra experiments.

2. All actuation (bending) is characterized during the cooling process, please add some data upon heating, just for comparison. If the deformation reversal becomes different or worse upon heating, that will be more interesting.

3. After spending quite some time on considering about the "central interest" for such deformation reversal, I have my own opinion: Line 234, the authors said, "stimulation on/off cycles may be reduced, which may simplify the control in many ways". I disagree. Operation of on/off light source is very simple, so, to double the response in the material is anyway more "expensive" than doubling the frequency of the light switch; The delay between bend and unbend events is due to the thermal response of the material, not easy to control by light, so it actually brings extra difficulty in precise robotic control.

However, I am with the authors on the application of multimodal locomotion by using deformation-reversal. This idea is not straightforward, because a reader may ask, a conventional LCN can possess multimodes by using different light intensity and pulses. But here, the key is that, as far as I would believe, every LCN robot requires cycling of deformation, such as bending forward and backward. When we talk about different "mode", different behaviors can be obtained just by playing different amplitude in one deformation cycle (conventional LCN), but it doesn't make any sense in "mode". The fact that, the LCN reported here can go across two deformation cycles upon one actuation, it does provide more options in obtaining actuation "mode". The concept is not very straightforward, neither the direct benefits bringing to soft robotic realization, but, I believe it is worthy to report and cause attention in the community.

Others:

1. The shape-change is defined by bending, and the measured angle is used to characterize such shape-change. I guess some "old-school" mechanics physicist won't agree. The "trick" here is, the actuator is 3D, the thickness and length are definitely changing during the actuation. But they are hidden, being not visible, while bending is the most visible effect in the observation. The point here is, the material matrix is continuously deforming in 3D, NOT just bend and unbending (one degree of freedom). Giving a few sentences to point out this will be appreciated.

2. Young's modulus of the material and how it changes upon temperature elevation? I think this is

important since the bending is due to the competition of elasticity between two surfaces. The situation in this work is more complicated: young's modulus is affected by the crosslinking density, and such density is not uniform across thickness; Stretching process included inner stress during the fabrication, which is then released to achieve actuation. So, no need to go deeply into these issues.

3. Please provide the UV-Vis spectra of the LCN, so one can calculate the 320 nm light penetration depth during the UV polymerization. This is important to understand the polymerization step, such as how deep the crosslinking process occurs inside material.

4. Line 254, please don't highlight "slipping" here. It is quite confusing in such a low-mass robot system without external force, what slipping means, maybe jumping, scratching on both sides alternatively, etc. When mentioning the slipping after the "at a speed ..." this would mislead the reader to the delusion that the robot may run very fast to slip for some distance.

5. slight error, line 457, ":", and some formatting inconsistency of "-"

6. Please add scale bars in all supporting figures. Especially the ones after stretching, it is hard to know the strip length

7. SI, Line 112. "The distance between light source (9.8 mW/mm²) and the sample is 9 cm." Since the authors have given information about light intensity, why this light source-sample distance matters? Or, is there any extra heating effect from the source, that is why they would like to mention the distance?

Dear Reviewers,

Thank you for your constructive comments and suggestions on our manuscript entitled “*Desynchronized Liquid Crystalline Network Actuators with Deformation Reversal Capability*” (manuscript NCOMMS- 20-38310). We are grateful to your recognizing the significance of this work. According to your comments and suggestions, we have made careful revisions on the original manuscript. All revised portions have been highlighted (both in the manuscript and the Supplementary Information). Our point-by-point response to your comments is given below.

Point-by-point response to reviewers' comments

(Reviewers comments in *italic*; revised/added texts in the manuscript highlighted in yellow)

Reviewer #1

The authors report a new type of liquid crystal network (LCN) actuator that deforms twice i.e. having three shapes upon applying one stimulus. This so-called deformation reversal has not been reported before and is interesting both from a fundamental and application point of view. Up to date only actuators have been described that have two shapes upon applying a single stimulus. The new actuators for example can be switched between a ring shape, a flat strip and a reverse ring shape of different size by changing only the temperature. The deformation reversal is programmed through asymmetrical crosslinking and/or asymmetrical stretching of the two sides of the LCN actuator. Furthermore, by using a photothermal coating, light can also be used to induce the shape changes. These novel findings lead to new unique possibilities in developing actuators and soft robot devices. I recommend publication of this well written manuscript after addressing the following issues.

1) It is not clear why an increasing crosslinking leads to widening of the LC-isotropic phase transition temperature range while the peak position does not change. The authors should investigate this behavior in more detail as this is the base of the desynchronized actuation.

Response :

We thank the reviewer for the positive comments and the recommendation of acceptance of this manuscript after revision. To address the first question, we have conducted more characterizations (Supplementary Fig. 1) and discussion of the phase transition behavior both in the manuscript and in the Supplementary Information.

The following discussion has been added in the revised manuscript (page 7):

This apparently similar peak maximum may be attributed to the following combined effects: 1) the effect of crosslinking the stretched sample in the LC phase which favors the macroscopic order and provides a tendency to increase the LC-isotropic phase transition temperature (T_{LC-I}), and 2) the effect of disordering the LC order by the crosslinks (acting as defects) which tends to reduce T_{LC-I} ^{44,45}. As for the broadening of the phase transition region, it can be caused by both polymer chain stretching and crosslinking. As previously reported⁴⁶, stretching of the used LCN can suppress the smectic phase in the polymer and emerge the initial smectic-nematic and nematic-isotropic phase transition peaks onto a broad nematic-isotropic peak. Furthermore, increasing the crosslinking degree of unstretched polymer appears to exert a similar effect due to the chain constraints that hinder the smectic layer formation (see more details in Supplementary Fig. 1).

In the Supplementary Information (page 2-4, added Supplementary Figure 1 and discussion):

This experiment was designed to observe the effect of only polymer chain crosslinking on the LC-isotropic phase transition of the used LCN. In order to obtain clear data on chain crosslinking density, photocrosslinking of the polymer dissolved in solution was carried out, which made it possible to monitor the increase in the photodimerization degree by ¹H NMR spectroscopy. The LCN samples dried from the solution, being unstretched and having various crosslinking densities, were used for the DSC measurements. The results clearly show the effect of chain crosslinking. Indeed, with increasing crosslinking density, the lower-temperature smectic-nematic phase transition peak in the uncrosslinked polymer is gradually weakened and suppressed to finally emerge with the nematic-isotropic transition peak. This leads to a broad nematic-isotropic phase transition and a broad thermomechanical or photomechanical actuation region (smectic-nematic transition does not provide a significant actuation). In the same time, the overall order-disorder phase transition enthalpy decreases with increasing the crosslinking. These results imply that higher chain crosslinking density impose greater chain topological constraints that have an effect to hinder the formation of smectic layer. This effect is similar to what was revealed in a previous study^{S1}: the mechanical stretching of the LCN can suppress the smectic ordering and give rise to a broad actuation temperature region.

The effect of chain crosslinking and chain stretching on LC behavior is a complex issue and depends on many factors (the used LCN, the order state of the sample during crosslinking, the crosslinking chemistry, polymer compositions and crosslinking agents of different chemical structures, to name a few).^{S2-S5} Nevertheless, in the present study, experimental evidence shows that both chain stretching and chain crosslinking can affect the order-disorder phase transition region. By applying different crosslinking densities and/or different chain stretching to the two sides of an LCN strip, the actuation temperature range as well as the resulting thermomechanical

or photomechanical forces on the two sides can be “designed” to differ from each other. This is the basis of the desynchronized actuator with deformation reversal capability.

2) The authors write that the performance of the actuators does not depend on thickness of the polymer film. However, the deformation should depend on the thickness of the low and high crosslinked layers. Do the authors have an explanation? Furthermore, what is the thickness of the two layers and is there also a region in the middle of the sample where the film is not crosslinked.

Response:

We thank the reviewer for this question. Previously, in our manuscript, we stated that the film thickness, stretching temperature and draw ratio “did not show a decisive effect on the deformation reversal behavior.” Through this, we wanted to emphasize the decisive effect of asymmetric crosslinking on the deformation reversal behavior. Considering that such statements might cause confusion and misunderstanding, we have revised this part of text as follows (revised manuscript, page 10-11):

Some other factors, such as film thickness, stretching temperature and draw ratio (λ , larger than 400%), have been investigated to further verify the decisive role of asymmetrical crosslinking in imparting the LCN actuator with the deformation reversal capability. When fixing the crosslinking times at 20 min and 120 min for the two sides of the strip, respectively, the deformation reversal behavior was observed for all samples investigated (including two LCN actuators prepared from linear polymer films with original thicknesses of 0.20 mm and 0.48 mm), although the performance of the actuator varies (Supplementary Fig. 7).

In addition, we have carried out new experiments in an effort to estimate the crosslinking depth of the LCN under the used conditions (added Supplementary Figures 11 and 12 and related discussion). We have also added a discussion in the manuscript as follows (page 11):

When a monolithic LCN film is subjected to asymmetrical photocrosslinking, its two sides not only differ in average crosslinking density but are likely to have different crosslinking depths along the thickness direction with a density gradient. Through the gel fraction measurement (Supplementary Fig. 11 and 12), the crosslinking depth was estimated to be between 0.010 mm to 0.071 mm with increasing the photocrosslinking time from 20 min to 120 min under the used UV light intensity (320 nm, 180 mW/cm²). However, it is hard to determine precisely the crosslinking depth and the crosslinking gradient. For a given dimension of the LCN strip, the deformation reversal behavior is governed by the competition between the lightly and highly crosslinked layers likely sandwiching an uncrosslinked layer in between. This competition could be a complex interplay of the strain difference, geometry change, mechanical properties (Supplementary Fig. 13) and internal stress, which are all temperature-dependent, leading to a range of bending angle changes as shown in Fig.

2e. On the basis of this principle, it can be expected that similar deformation reversal behavior could be obtained for LCNs with different thicknesses, but the necessary photocrosslinking times vary.

3) *The reversibility and fatigue of the actuators is not clear. In most cases only cooling data is presented (for example, Figure 2, 3). The actuation behavior upon heating should also be given. How many times can the actuation be repeated?*

Response:

We have provided data to show the actuation behavior on heating (please see Supplementary Fig. 2 and the new Movie 1). Cyclic actuation was also performed to demonstrate the stability of the actuation behavior (see Supplementary Fig. 6). Furthermore, we want to point out that all the photothermally driven actuators, shown in Fig. 4, 5 and Supplementary Fig. 20, 21, show the reversible deformation reversal behavior on heating followed by cooling.

Related revision in the manuscript (page 9 and 10):

On subsequent heating, the actuator executes the opposite deformation reversal steps (Supplementary Fig. 2 and Movie 1).

As the deformation reversal capability is structurally-inscribed in the LCNs via tuning the crosslinking times, the resulting actuator is stable and displays the deformation reversal under repeated heating/cooling cycles (20 cycles monitored, Supplementary Fig. 6). Although the recorded $|\Delta\alpha_I|$ and $|\Delta\alpha_{II}|$ decreased slightly, which could be attributed to some unrestored order during actuation under the used conditions, the initial performance can be recovered by simply setting the actuator in air at room temperature for a couple of hours.

4) *The characterization of the polydopamine actuators is poor. In the experimental only the preparation procedure is given.*

Response:

We thank the reviewer for this suggestion. We have added the UV-Vis, FTIR and SEM characterizations of the polydopamine actuators (see Supplementary Fig. 16 and 17).

Reviewer #2

In this manuscript entitled “Desynchronized Liquid Crystalline Network Actuators with Deformation Reversal Capability”, the authors provided a desynchronized actuation strategy by endowing the two sides of a monolithic LCN with different crosslinking

densities or by crosslinking the two sides at different elongation strain, to prepare a novel type of LCN, which could reversibly deform among three different pre-set shapes on heating or under light irradiation. The most significant contribution of this work is that (1) it realizes the multi-stage shape changes of a monolithic LCN film in one stimulation on/off cycle; (2) the asymmetrical crosslinking methods can be potentially extended to various actuator systems to design and manufacture soft robots which could complete complex motions under the control of a simple external stimulus. The paper is well organized and written. The conclusions are well supported by the convincing data and characterization. Overall, I believe the work will attract interest of a broad readership and will have far reaching impact in LCN research field. Therefore, I strongly recommend publication of this manuscript after the following minor revisions (listed below) are addressed:

Questions:

1. Scale bars should be added in the photo images of the actuators in Figure 4 and Figure 5.

Response:

We are grateful to the reviewer for the positive evaluation of our work and the recommendation of accepting the manuscript after revision. As requested by the reviewer, we have added scale bars in the photos in Figure 4 and Figure 5.

2. The penetration of UV light is relatively poor, which might cause a gradient of crosslinking density along the thickness direction of the LCN film. What is the relationship between the thickness of the LCN films and their responsive behaviors?

Response:

Thanks for the question. Indeed, there should be a gradient of crosslinking density along the thickness direction of the LCN film. Under fixed crosslinking conditions, varying the thickness of the LCN films can result in different deformation reversal behaviors due to the geometric effect and the relative change of the crosslinking gradient with respect to the entire cross-section of the LCN. We have added a discussion about this point in the manuscript. A short answer is that the optimal crosslinking times for deformation reversal should be shifted depending on the thickness of the film.

Related revision in the revised manuscript (page 11):

When a monolithic LCN film is subjected to asymmetrical photocrosslinking, its two sides not only differ in average crosslinking density but are likely to have different crosslinking depths along the thickness direction with a density gradient. Through the gel fraction measurement (Supplementary Fig. 11 and 12), the crosslinking depth was estimated to be between 0.010 mm to 0.071 mm with increasing the photocrosslinking time from 20 min to 120 min under the used UV light intensity (320 nm, 180 mW/cm²). However, it is hard to determine precisely the crosslinking depth and the crosslinking gradient. For a given dimension of the LCN strip, the deformation reversal behavior is governed by the competition between the lightly and highly crosslinked layers likely sandwiching an uncrosslinked layer in between. This

competition could be a complex interplay of the strain difference, geometry change, mechanical properties (Supplementary Fig. 13) and internal stress, which are all temperature-dependent, leading to a range of bending angle changes as shown in Fig. 2e. On the basis of this principle, it can be expected that similar deformation reversal behavior could be obtained for LCNs with different thicknesses, but the necessary photocrosslinking times vary.

3. Both the phrases “liquid crystalline networks” and “liquid crystal networks” are used in this manuscript. Although both phrases are used by researchers in this field, it is better for the authors to unify them in order not to cause confusion in more extensive audience.

Response:

Thanks for the suggestion. We have revised these phrases in the manuscript using “liquid crystalline networks”.

4. What if we prepared a fiber using the strategy of this work? Could the multi-stage deformations be realized?

Response:

This is a great question. Actually we tried to make a fiber capable of deformation reversal by manually stretching a cylindrical sample (the length and the diameter of the stretched sample are about 18 mm and 0.2 mm respectively) and crosslinking the two parts of it for 20 min and 120 min respectively. However, due to the heterogeneity induced during processing, the fiber twisted while bending/unbending. Therefore, we would say that this mechanism is also applicable for fibers, but more technical details need to be improved to handle fibers more accurately, especially for very thin ones. The result is not provided in this paper, but this is a very interesting issue and worth being explored in our future work.

Reviewer #3

The paper “Desynchronized Liquid Crystalline Network Actuators with Deformation Reversal Capability” by Yue Zhao, et al, reported a fabrication of asymmetric liquid crystal network actuator that can exhibit bending and reversed bending upon heat (or light) stimulus. The deformation reversal is achieved by tuning the phase transition temperatures between two sides, rendering a so called “desynchronized” process. Detailed experiments (controlling the polymerization time and stretching) are applied to control such desynchronized actuation behaviors. Applications in multi-modal walking robot are demonstrated. The story is well written, providing certain advances to the field.

Hopefully the following comments could help to further improve the quality.

The concept in general.

I am a bit skeptical about the novelty level raised by the authors. I think it is inappropriate to make a strong point at the beginning, by saying, "peculiar and fascinating actuation behavior", "has not been conceptualized and explored before". i) If an actuator is constructed with two sides of different materials that contract at different temperatures, it should firstly bend and then unbend. This phenomenon is quite obvious, at least to me. ii) There are some similar demonstrations reported before, as the authors also mentioned at the very last section of the manuscript (line 350).

So, my suggestions are,

i) Bring Ref. 44-46, the studies about bend-unbent events in LCN, to the introduction for comparison. The authors can still claim they are "the first one", but, to highlight such deformation reversal mechanism and use it in soft robotic purposes. The solid scientific pieces in this manuscript are the fact that, the authors have developed comprehensive experimental procedure by changing the polymerization time and stretching ratio, to obtain a well-control of such deformation reversal.

ii) What really distinguish the deformation reversal reported in this work from those in Ref.44-46 is that, the bending (at certain temperature) is obtained/measured at thermal equilibrium. Many thermal or photo responsive LCN actuators, may exhibit bending-to-unbending deformation upon one stimulus, only because they are moving toward the thermal equilibrium (in the case of change of thermal gradient, or say, when the heat is going from one surface to another), or photo stationary state (in the case of cis azo population across the thickness). Those bending-unbending deformation are just intermediate states during the time span of response evolution. I believe such difference is significant, however, not easy to explain in few words.

iii) Line 43. the authors mentioned in the introduction, the typical way of shape change is from shape1 to shape2. I understand and also agree with the authors on this. But, readers may confuse with the reconfigurability concept which has been widely reported in LCNs, since they can produce different shapes upon the same stimulus.

Now, the introduction is very short, I strongly recommend the authors to twist the points listed above, and reshape the introduction. After that, the novelty should be more clear, and easier for readers to appreciate.

Response:

We thank the reviewer for careful/insightful analysis and detailed suggestions. We have thoroughly revised and reconstructed the introduction by taking into consideration the points raised above. Please see pages 3 and 4 in the revised manuscript.

Main points:

1. *All the deformations are characterized by measurement of the bending angle. However, the bending angle of a long strip is so sensitive to the thickness and length. All the sample are prepared with 350 micron in thick, BUT, after the stretching, thickness and length both change. This actually brings some hurdles and complexity in comparing data (Fig.3b). Although these parameters won't bring a "decisive" effect as the authors mentioned, they*

do affect the bending angle characterization, especially in the experiments for a well-control of such behaviors (Fig.2e). I would suggest the author to provide thickness, length data of each actuator, and more discussion on this. It is unrealistic to repeat the all experiments by using sample with same thickness. So, this comment doesn't require extra experiments.

Response:

Thanks for raising this point. As suggested, we have now provided the thickness and length of all samples used in the different experiments in Supplementary Table 1 (Supplementary Information, page 29). We have also added discussion in the revised manuscript (page 9 and 14):

Since the bending angle is related to the geometry of the specimen⁴⁹, for a bending angle comparison to be meaningful in characterizing the deformation reversal behavior, we kept similar length and thickness for the different LCN actuators (see Supplementary Table 1).

It should be noted that the slight differences in length and thickness of these samples (see Supplementary Table 1) can cause a certain variation in the observed bending angles, but the overall trend is not affected.

2. All actuation (bending) is characterized during cooling process, please add some data upon heating, just for comparison. If the deformation reversal becomes different or worse upon heating, that will be more interesting.

Response:

Thanks for this suggestion. We have added Supplementary Fig. 2 and the revised Movie 1 to show the actuation behavior upon heating. The total bending angle change during heating is slightly lower than that during cooling, but in general, the deformation reversal behaviors in both the heating and cooling processes are quite similar.

Related revision in the manuscript (page 9):

On subsequent heating, the actuator executes the opposite deformation reversal steps (Supplementary Fig. 2 and Movie 1).

3. After spending quite some time on considering about the "central interest" for such deformation reversal, I have my own opinion: Line 234, the authors said, "stimulation on/off cycles may be reduced, which may simplify the control in many ways". I disagree. Operation of on/off light source is very simple, so, to double the response in the material is anyway more "expensive" than doubling the frequency of the light switch; The delay between bend and unbend events is due to the thermal response of the material, not easy to control by light, so it actually brings extra difficult in precise robotic control.

However, I am with authors on the application of multimodal locomotion by using deformation-reversal. This idea is not straightforward, because a reader may ask, a conventional LCN can possess multimodes by using different light intensity and pulses. But here, the key is that, as far as I would believe, every LCN robot requires cycling of

deformation, such as bending forward and backward. When we talk about different “mode”, different behaviors can be obtained just by playing different amplitude in one deformation cycle (conventional LCN), but it doesn’t make any sense in “mode”. The fact that, the LCN reported here can go across two deformation cycles upon one actuation, it does provide more options in obtaining actuation “mode”. The concept is not very straightforward, neither the direct benefits bringing to soft robotic realization, but, I believe it is worthy to report and cause attention in the community.

Response:

We are grateful for the reviewer’s thoughtful analysis and valuable insights on the possible benefits of the deformation-reversal actuators. We have revised the related texts in the manuscript, including the removal of “which may simplify the control in many ways” and the addition of a discussion that specifies the meaning of different modes.

Page 16:

The central interest of the deformation reversal resides in the fact that the number of bidirectional shape switch can be doubled under one stimulation on/off cycle. Since repeated shape switch under repeated on/off stimulation is the basis of moving LCN soft robots, this feature can be explored.

Page 20:

As a result, different movement fashions, driven by side-on out-of-plane bending/unbending and/or edge-on in-plane bending/unbending, can readily be achieved using the same LCN actuator under light exposure of constant intensity. To simply the discussion, these different locomotion patterns involving different types of actuator deformation will be referred to as different modes.

Others:

1. The shape-change is defined by bending, and the measured angle is used to characterize such shape-change. I guess some “old-school” mechanics physicist won’t agree. The “trick” here is, the actuator is 3D, the thickness and length are definitely changing during the actuation. But they are hidden, being not visible, while bending is the most visible effect in the observation. The point here is, the material matrix is continuously deforming in 3D, NOT just bend and unbending (one degree of freedom). Giving few sentences to point out this will be appreciated.

Response:

Thanks for this remark. We have added a discussion to point out this in the manuscript. Related revision in the manuscript (page 10):

A remark must be made here. The shape change of LCN actuators is often characterized using bending/unbending, simply because this is the most visible deformation. The actual shape change is a continuous 3D deformation of the soft LCN

material matrix (i.e., change in geometry, length, thickness and width).

2. *Young's modulus of the material and how it changes upon temperature elevation? I think this is important since the bending is due to the competition of elasticity between two surfaces. The situation in this work is more complicated: young's modulus is affected by the crosslinking density, and such density is not uniform across thickness; Stretching process included inner stress during the fabrication, which is then released to achieve actuation. So, no need to go deeply into these issues.*

Response:

Thanks for this discussion. Indeed, the bending behavior is a complex interplay among the modulus, sample size, strain difference, internal stress and the crosslinking/mechanical strength gradients. It is hard to elucidate the role of each factor in determining the bending behavior. Nevertheless, we have performed some new experiments and added the Young's modulus of the LCN films with different photocrosslinking times measured at three different temperatures. Please see Supplementary Fig. 13.

3. *Please provide the UV-Vis spectra of the LCN, so one can calculate the 320 nm light penetration depth during the UV polymerization. This is important to understand the polymerization step, such as how deep the crosslinking process occurs inside material.*

Response:

Thanks for this suggestion. We have provided a UV-Vis spectrum of the LCP spin-coated on a quartz plate (Supplementary Figure 16). However, the UV-vis spectra of the actual LCN actuators (much thicker) under the used UV light irradiation cannot be obtained. To evaluate the crosslinking depth, we have carried out new experiments, added data in Supplementary Figures 11 and 12 and made discussion.

Related revision in the manuscript (page 11):

When a monolithic LCN film is subjected to asymmetrical photocrosslinking, its two sides not only differ in average crosslinking density but are likely to have different crosslinking depths along the thickness direction with a density gradient. Through the gel fraction measurement (Supplementary Fig. 11 and 12), the crosslinking depth was estimated to be between 0.010 mm to 0.071 mm with increasing the photocrosslinking time from 20 min to 120 min under the used UV light intensity (320 nm, 180 mW/cm²). However, it is hard to determine precisely the crosslinking depth and the crosslinking gradient. For a given dimension of the LCN strip, the deformation reversal behavior is governed by the competition between the lightly and highly crosslinked layers likely sandwiching an uncrosslinked layer in between. This competition could be a complex interplay of the strain difference, geometry change, mechanical properties (Supplementary Fig. 13) and internal stress, which are all temperature-dependent, leading to a range of bending angle changes as shown in Fig. 2e. On the basis of this principle, it can be expected that similar deformation reversal behavior could be obtained for LCNs with different thicknesses, but the necessary photocrosslinking times vary.

4. Line 254, please don't highlight "slipping" here. It is quite confusing in such a low-mass robot system without external force, what slipping means, maybe jumping, scratching on both sides alternatively, etc. When mentioning the slipping after the "at a speed ..." this would mislead the reader to the delusion that the robot may run very fast to slip for some distance.

Response

Thanks for pointing this out. We have removed this part from the manuscript.

5. slight error, line 457, ":", and some formatting inconsistency of "-"

Response:

We have corrected these errors in the manuscript.

6. Please add scale bars in all supporting figures. Especially the ones after stretching, it is hard to know the strip length.

Response:

Thanks for the suggestion. We have added the scale bars in all supporting figures, and provided a table (Supplementary Table 1) showing the dimensions of all actuators used in this work.

7. SI, Line 112. "The distance between light source (9.8 mW/mm^2) and the sample is 9 cm." Since the authors have given information about light intensity, why this light source-sample distance matters? Or, is there any extra heating effect from the source, that is why they would like to mention the distance?

Response:

The distance doesn't really matter, it was mentioned just for more experimental details. Even though there may be some heating from the light source, the photothermal effect due to the polydopamine layer plays the main role in heating. We have removed the distance to avoid any possible confusion.

REVIEWERS' COMMENTS

Reviewer #1 (Remarks to the Author):

The authors have addressed the comments adequately.
The manuscript is now suitable for publication.

Reviewer #2 (Remarks to the Author):

The authors have successfully addressed my previous concerns. The paper is recommended for publication.

Reviewer #3 (Remarks to the Author):

My opinion is that, the authors have done a significant addition to their previous manuscript, addressing all the reviewers' comments in full, particularly the issues of crosslinking gradient, absorption depth, thickness, etc.
The manuscript is ready to go.

Some side notes, no need for an extra review round.

1. It seems, according to Nature Communication's policy, one needs to separate supporting figures and discussions.
2. English in the supplementary Discussion lose a little bit, e.g., line 55, "the used LCN", old LCN, material in use, or at different actuation cycles?
3. In one of the comments about fiber actuator, the authors mentioned, "However, due to the heterogeneity induced during processing, the fiber twisted while bending/unbending." If the heterogeneity can cause symmetry breaking (which is also common in soft matters), and can switch from left handed twisting to right handed upon light actuation, it will be of significant.

Point-by-point response to reviewers' comments

(Reviewers' comments in italic; responses in blue)

Reviewer #1

The authors have addressed the comments adequately.

The manuscript is now suitable for publication.

Response: We appreciate the reviewer for recommending publication of the revised manuscript.

Reviewer #2

The authors have successfully addressed my previous concerns. The paper is recommended for publication.

Response: We are grateful to the reviewer for recommending publication of the revised manuscript .

Reviewer #3

My opinion is that, the authors have done a significant addition to their previous manuscript, addressing all the reviewers' comments in full, particularly the issues of crosslinking gradient, absorption depth, thickness, etc.

The manuscript is ready to go.

Response: We thank the reviewer for the approval of our revision.

Some side notes, no need for an extra review round.

1. It seems, according to Nature Communication's policy, one needs to separate supporting figures and discussions.

Response: We thank the reviewer for this reminder. We have separated the supporting figures and discussions in Supplementary information.

2. English in the supplementary Discussion lose a little bit, e.g., line 55, "the used LCN", old LCN, material in use, or at different actuation cycles?

Response: We thank the reviewer for pointing this out. We have changed "the used LCN" to "the LCN used" (meaning material in use, see line 3 and line 20 on page 26 in Supplementary information).

3. In one of the comments about fiber actuator, the authors mentioned, "However, due to the heterogeneity induced during processing, the fiber twisted while bending/unbending." If the heterogeneity can cause symmetry breaking (which is also common in soft matters), and can switch from left handed twisting to right handed upon light actuation, it will be of significant.

Response: We thank the reviewer for this interesting point. We believe the switch from left handed to right handed twisting upon light actuation is possible and will continue to study this in the follow-up work.